# Establishing charge-transfer excitons in 2D perovskite heterostructures

Jia Zhang [1,2], Xixiang Zhu[1,2], Miaosheng Wang [2] & Bin Hu [2✉]

Charge-transfer excitons (CTEs) immensely enrich property-tuning capabilities of semiconducting materials. However, such concept has been remaining as unexplored topic within halide perovskite structures. Here, we report that CTEs can be effectively formed in heterostructured 2D perovskites prepared by mixing $PEA_2PbI_4$:$PEA_2SnI_4$, functioning as host and guest components. Remarkably, a broad emission can be demonstrated with quick formation of 3 ps but prolonged lifetime of ~0.5 μs. This broad PL presents the hypothesis of CTEs, verified by the exclusion of lattice distortion and doping effects through demonstrating double-layered $PEA_2PbI_4$/$PEA_2SnI_4$ heterostructure when shearing-away $PEA_2SnI_4$ film onto the surface of $PEA_2PbI_4$ film by using hand-finger pressing method. The below-bandgap photocurrent indicates that CTEs are vital states formed at $PEA_2PbI_4$:$PEA_2SnI_4$ interfaces in 2D perovskite heterostructures. Electroluminescence shows that CTEs can be directly formed with electrically injected carriers in perovskite LEDs. Clearly, the CTEs presents a new mechanism to advance the multifunctionalities in 2D perovskites.

[1] Key Laboratory of Luminescence and Optical Information Ministry of Education, Beijing Jiaotong University, 100044 Beijing, China. [2] Department of Materials Science and Engineering, University of Tennessee, Knoxville, TN 37996, USA. ✉email: bhu@utk.edu

nitiated by fast-developing thin-film photovoltaic functional-ities[1–5], organic–inorganic metal halide hybrid perovskites have emerged as interesting light-emitting materials with high pho-toluminescence (PL) quantum yields[6–8], large color tuning cap-abilities[9–11], domain-size controllable exciton binding energies[12,13] and bandgap engineering properties[14–16]. With these attractive optical characteristics, the hybrid perovskites have demonstrated high-performance light-emitting diodes (LEDs)[7,17–20] based on solution-processed thin-film formation procedures. Undoubtedly, the intrinsic excitons formed within band structures function as the primary excited states responsible for developing high-performance light-emitting properties in such hybrid perovskites. Essentially, tuning the intrinsic excitons through the formation probability, binding energy, and radiative/nonradiative recombi-nation can determine the light-emitting properties, as exampled by mixing different halides[9–11], introducing nanostructures[15,21,22], and passivating grain boundary defects[18,23,24]. On the other hand, it is interesting to note that CTEs have been introduced as artifi-cially engineered excitons with delocalized wavefunctions, namely, spatially extended states, and widely observed in low-dielectric-constant organic materials through donor–acceptor design, to develop high-efficiency optoelectronic functionalities[25–29]. It has been recognized as an intriguing nature that CTEs can be con-veniently tuned on energy, polarization, and spin parameters by physically changing the formulation of donor–acceptor heterostructures[30,31]. This leads to unique tuning capabilities to control optic, electric, and magnetic functionalities by using CTEs[32–35]. In general, CTEs can be formed at heterostructures where charge transfer occurs between different local structures to form electron–hole pairs[36,37]. The heterostructures can be both chemically and physically prepared by using two molecular structures where different local energies and electron negativities exist[38,39]. In general, the heterostructures can be naturally for-mulated between donor- and acceptor-type structures, exampled as highly efficient light-emitting exciplexes[40–42]. On the other hand, the heterostructures can be also formed between two same-type molecular structures where local energy disorders are occurred, shown as light-emitting excimers in organic materials[43,44]. In the past, the CTEs have been extensively explored in organic hetero-structures prepared by chemical synthesis, physical mixing, and mechanical formation of film interfaces in light-emitting[10,11,45], lasing[46,47], photovoltaic[38,39], sensing[48], and magneto-optical[49] applications. Recently, it has been shown that the perovskite–polymer bulk heterostructure, prepared by mixing quasi-2D/3D perovskite with wide-bandgap polymer, demonstrates a broad transient absorption (TA) signal, which is quickly appeared in 1 ps and slowly relaxed in nanoseconds[19]. This presents the possibility to form CTEs between perovskite and organic polymer, known as high- and low-dielectric materials. Essentially, the CTEs can be formed when a charge-transfer process is occurred with the con-sequence of forming electron–hole pairs between two adjacent structures with different electron negativities and local energies. It has also been observed that CTEs can be formed between 2D perovskite ($BA_2PbI_4$) and organic molecule (1,4,5,8,9,11-hexaaza-triphenylenehexacarbonitrile) when the charge transfer at the interface is enhanced by orbital overlap[50]. Clearly, these published results provide the necessary condition to explore the CTEs within perovskite heterostructures.

In this work, we initially utilized the chemical method to prepare 2D perovskite heterostructures $[(PEA_2PbI_4)_x:(PEA_2SnI_4)_{1−x}]$ by alternatively selecting the host and guest components when mixing two precursor (Pb and Sn per-ovskites) solutions with distinctly high and low concentrations. The heterostructures with alternatively switched host and guest components are both demonstrated with similar broad emission spectra and prolonged lifetime (~microseconds).

This observation provides the first indication that the CTEs are formed at the $[PEA_2PbI_4:PEA_2SnI_4]$ interfaces within heterostructures $[(PEA_2PbI_4)_{0.999}:(PEA_2SnI_4)_{0.001}]$ and $[(PEA_2SnI_4)_{0.999}:(PEA_2PbI_4)_{0.001}]$ to generate a broad emission in 2D perovskites. Remarkably, the broad emission can be reproduced when using hand–finger pressing method to mechanically shearing away the $PEA_2SnI_4$ film onto the $PEA_2PbI_4$ film surface to uniquely prepare the double-layered heterostructures without lattice distortion and doping effects. This hand–finger pressing method confirms that the broad emission represents the signature for CTEs formed in 2D per-ovskite heterostructures. The pump–probe TA studies present that the CTEs are quickly formed and essentially become metastable states, as compared to surface-trapped and intrinsic excitons in 2D perovskites.

## Results

**CTEs in heterostructured 2D perovskite thin film.** The 2D perovskite films were prepared by spin coating from mixed two precursor solutions ($PEA_2PbI_4$ and $PEA_2SnI_4$) and followed with thermal annealing at 100 °C for 10 min. The two precursor solutions were initially mixed with the volume ratio of 99.9:0.1 to formulate the heterostructures $[(PEA_2PbI_4)_{0.999}:(PEA_2SnI_4)_{0.001}]$ in 2D perovskite films where the $PEA_2PbI_4$ and $PEA_2SnI_4$ act as host and guest components, respectively. Furthermore, the $PEA_2PbI_4$ and $PEA_2SnI_4$ were alternatively switched with distinctly low and high concentra-tions, serving as guest and host components, to formulate the het-erostructures $[(PEA_2SnI_4)_{0.999}:(PEA_2PbI_4)_{0.001}]$ in 2D perovskite films. The X-ray diffraction (XRD) data in Supplementary Fig. 1 indicate that all the 2D perovskite films ($PEA_2PbI_4$, $[(PEA_2PbI_4)_{0.99}:(PEA_2SnI_4)_{0.01}]$, $[(PEA_2PbI_4)_{0.95}:(PEA_2SnI_4)_{0.05}]$, $[(PEA_2PbI_4)_{0.05}:(PEA_2SnI_4)_{0.95}]$, and $PEA_2SnI_4$) show excellent crystallinity with extremely narrow dominant peak at [0 0 2] direction. Furthermore, the peak of $PEA_2SnI_4$ film is slightly redshifted as compared with the $PEA_2PbI_4$ film due to smaller crystalline size. By increasing the ratio of the guest ($PEA_2SnI_4$) component in 2D perovskite hetero-structured $[(PEA_2PbI_4)_{0.95}:(PEA_2SnI_4)_{0.05}]$ film, the XRD peak is redshifted with decreased intensity towards that of pristine $PEA_2SnI_4$ film. This provides an evidence that, in the 2D perovskite hetero-structured film, both $PEA_2PbI_4$ and $PEA_2SnI_4$ perovskites are formed. The scanning electron microscopy and atomic force microscopy results show that all the films are of high quality (Sup-plementary Figs. 2 and 3). Figure 1a shows that mixing 0.1% $PEA_2SnI_4$ into 99.9% $PEA_2PbI_4$ has little effect on the absorption characteristics in the heterostructures $[(PEA_2PbI_4)_{0.999}:(PEA_2SnI_4)_{0.001}]$ where the bandgap is governed by the host ($PEA_2PbI_4$) component. Interestingly, a very broad PL spectrum is observed between 575 and 800 nm with the peak position located at 669 nm. When we further increase the guest Sn perovskite con-centration to 1% or even 5%, broad emission can always be observed at same wavelength (Supplementary Fig. 4). As a comparison, the $PEA_2PbI_4$ and $PEA_2SnI_4$ perovskites give rise to their intrinsic PL at 524 and 621 nm, shown as spectral shoulders on the broad emission spectrum. The broad PL demonstrates a very broad spectral width of 127 nm, while the intrinsic emission from the $PEA_2PbI_4$ and $PEA_2SnI_4$ components give rise to much narrower spectral widths (15 and 37 nm). Remarkably, when the host and guest components are alternatively switched to shift the bandgap (derived from Sup-plementary Fig. 5) largely from 2.33 eV in the $[(PEA_2PbI_4)_{0.999}:(PEA_2SnI_4)_{0.001}]$ to 1.95 eV in the $[(PEA_2SnI_4)_{0.999}:(PEA_2PbI_4)_{0.001}]$, the broad emission spectrum is only slightly changed from 669 nm (1.85 eV) to 697 nm (1.78 eV) in peak position, leading to similar broad emission spectra between $[(PEA_2PbI_4)_{0.999}:(PEA_2SnI_4)_{0.001}]$ and $[(PEA_2SnI_4)_{0.999}:(PEA_2PbI_4)_{0.001}]$ heterostructures, as indicated in Fig. 1b. The similar broad emission observed by alternatively

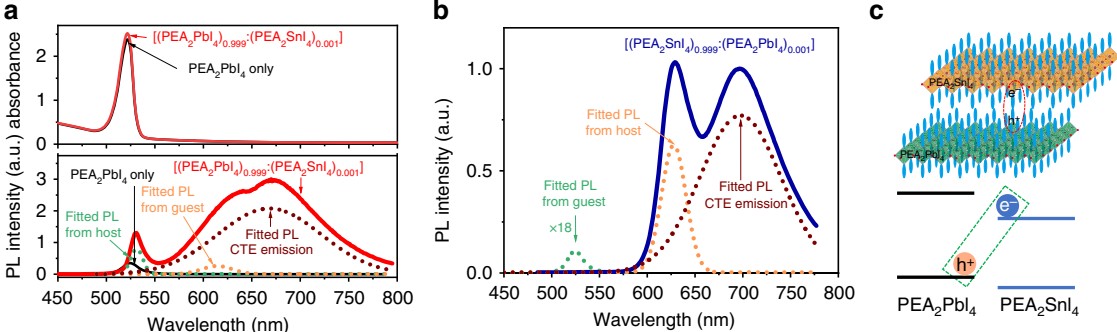

**Fig. 1 Optical characteristics of CTE formation in 2D perovskite heterostructures. a** Absorption and PL spectra are shown for 2D perovskite heterostructured $[(PEA_2PbI_4)_{0.999}:(PEA_2SnI_4)_{0.001}]$ film. The absorption spectrum indicates that the bandgap is governed by the host ($PEA_2PbI_4$) component in the heterostructures. The PL spectrum is dominated by a broad emission peaked at 669 nm with a spectral shoulder at 614 nm from the guest ($PEA_2SnI_4$) and the intrinsic emission (peaked at 531 nm) from the host ($PEA_2PbI_4$). **b** PL spectrum is shown for 2D perovskite heterostructured $[(PEA_2SnI_4)_{0.999}:(PEA_2PbI_4)_{0.001}]$ film after switching host and guest components. The broad emission is observed with the peak position at 697 nm, in addition to intrinsic emission (peaked at 524 and 627 nm) from host ($PEA_2PbI_4$) and guest ($PEA_2SnI_4$) components. The PL spectra for pristine $PEA_2PbI_4$ and $PEA_2SnI_4$ films are also shown as reference. **c** Schematic diagram illustrates the formation of CTEs at the interfaces between $PEA_2PbI_4$ and $PEA_2SnI_4$ components in 2D perovskite heterostructures $[(PEA_2PbI_4)_{1-x}:(PEA_2SnI_4)_x]$.

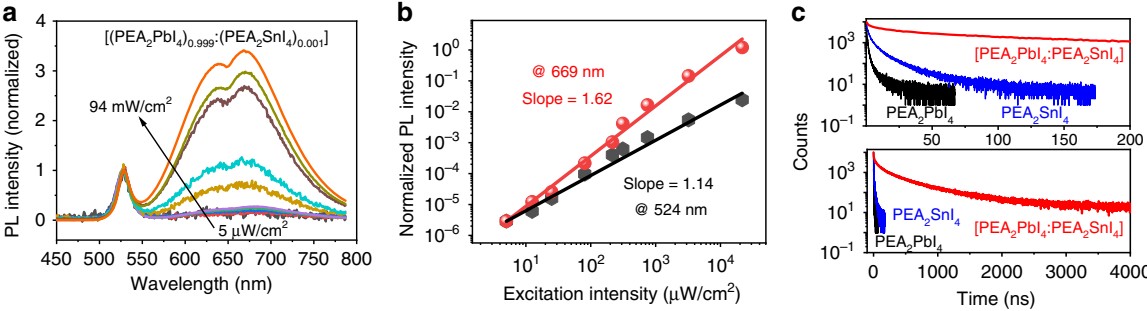

**Fig. 2 PL characteristics for 2D perovskite heterostructured $[(PEA_2PbI_4)_{0.999}:(PEA_2SnI_4)_{0.001}]$ film. a** Normalized PL spectra at different excitation intensities ranging from 5 to 94 mW/cm². **b** PL-excitation intensity dependences for broad emission (peaked at 669 nm) as compared to intrinsic emission (524 nm) from host ($PEA_2PbI_4$). **c** PL lifetime results for broad emission (peaked at 669 nm) in 2D perovskite heterostructures and intrinsic emission (peaked at 524 nm, and 621 nm) from host ($PEA_2PbI_4$) and guest ($PEA_2SnI_4$).

switching the host and guest components in 2D perovskite heterostructures $[(PEA_2PbI_4)_{0.999}:(PEA_2SnI_4)_{0.001}]$ and $[(PEA_2SnI_4)_{0.999}:(PEA_2PbI_4)_{0.001}]$ lead to the hypothesis that the CTEs are formed at the interfaces between local $PEA_2PbI_4$ and $PEA_2SnI_4$ structures, as illustrated in Fig. 1c. Here, the formation of CTEs requires that (a) the charge transfer is occurred from local host to guest structures and (b) the electron–hole pairs are formed at the interfaces in the 2D perovskite heterostructures.

To further understand the origin of broad emission occurring in the 2D perovskite heterostructures, we measured the PL intensity-dependence and lifetime characteristics. We can see in Fig. 2a, b that the broad emission exhibits the intensity-dependence slope of 1.62, while the intrinsic emission from the host $PEA_2PbI_4$ component gives rise to the intensity-dependence slope of 1.14. We should note that germinate process dominates the radiative recombination of the intrinsic excitons in host/guest 2D perovskites due to large exciton binding energy, which leads to the power-dependence slope of ~1. On the other hand, when electrons and holes are paired to form electron–hole pairs, the pairing probability $p$ is proportional to the product between the number ($n$) of electrons and the number ($m$) of holes: $p \propto m \cdot n$. In the situation where $n = m$, then $p \propto n^2$. Therefore, the non-germinate recombination can lead to the intensity-dependence slope of 2 in light emission. Here, the broad emission of our heterostructured $[(PEA_2PbI_4)_{0.95}:(PEA_2SnI_4)_{0.05}]$ film demonstrates the power-dependence slope of 1.62, presenting the suggestion

that the CTEs are responsible for the broad emission through non-germinate recombination. Furthermore, the broad emission peaked at 669 nm shows much prolonged lifetime of 0.5 μs relative to the short lifetime of 1 ns of the intrinsic emission peaked at 524 nm from the host $PEA_2PbI_4$ component, as shown in Fig. 2c. Detailed fittings of PL lifetimes of pure $PEA_2PbI_4$ and $PEA_2SnI_4$ films are shown in Supplementary Fig. 6. In general, the lifetime of CTEs is mainly determined by the electron–hole recombination rate governed by Coulomb attractive force. As a result, CTEs often exhibit an extended lifetime due to longer electron–hole separation distance as compared to intrinsic excitons in semiconducting materials. Here, the much prolonged PL lifetime further verifies that the broad emission originates from CTEs in heterostructured 2D perovskites. We should note that CTEs can still demonstrate various lifetimes when the Coulomb attraction between electron and hole located on different energetic structures is changed by local dielectric backgrounds, widely observed in organic–organic[51–53], organic–inorganic[35,54], and inorganic–inorganic mixtures[34,55].

**CTEs in double-layered 2D perovskite heterostructures**. We note that, in the mixed two precursor solutions with 99.9% host and 0.1% guest concentrations, the $PEA_2PbI_4$ and $PEA_2SnI_4$ have slower and faster crystallization rates to form heterostructured 2D perovskite $[(PEA_2PbI_4)_{0.999}:(PEA_2SnI_4)_{0.001}]$ films. This is because the Pb and Sn perovskites have higher and lower Gibbs energies,

respectively[56]. Consequently, the $PEA_2SnI_4$ domains can be quickly formed and followed with the formation of $PEA_2PbI_4$ domains, leading to the heterostructured 2D perovskite films. The heterostructures provide the necessary condition to initiate the charge transfer from the host to the guest domains towards the formation of CTEs at the host:guest interfaces. Essentially, the CTEs can be formed through Coulomb capture at the interfaces between $PEA_2PbI_4$ and $PEA_2SnI_4$ domains in 2D perovskite heterostructures. However, there is a question whether the low-concentration guest ($PEA_2SnI_4$) molecules dope into the high-concentration host ($PEA_2PbI_4$) domains, leading to the possibility of doping effect to generate a broad emission through the so-called self-trapped excitons caused by lattice deformation, which was proposed for doping-induced deformation in (1 0 0) 2D single crystals[57] and corrugated (1 1 0) 2D structures[58,59]. To clarify this question, we designed a unique experimental method to mechanically prepare the double-layered $PEA_2PbI_4/PEA_2SnI_4$ heterointerface by shearing away the $PEA_2SnI_4$ film onto the surface of $PEA_2PbI_4$ film through hand–finger pressing procedure. Specifically, the $PEA_2SnI_4$ and $PEA_2PbI_4$ films were separately spin cast on glass substrates, followed by thermal annealing. Then, the $PEA_2SnI_4$ film coated on the glass substrate was made to hold onto the $PEA_2PbI_4$ film coated on another glass substrate with face-to-face contact under hand–finger pressing, as schematically illustrated in Fig. 3a. Upon hand–finger pressing with face-to-face contact, the $PEA_2SnI_4$ film was parallelly shearing away from the surface of $PEA_2PbI_4$ film. This hand–finger pressing method leaves the $PEA_2SnI_4$ film islands on the surface of $PEA_2PbI_4$ film (see the microscopic photos in Supplementary Fig. 7), forming the double-layered $[PEA_2PbI_4/PEA_2SnI_4]$ interfaces, functioning as heterostructures clearly without doping effects and lattice deformation. Therefore, our hand–finger pressing method provides a unique experimental approach to clarify the origin of broad emission widely observed in hybrid metal halide perovskite heterostructures formed by mixing different precursor solutions or in nanocrystals formed with lattice strains. Here, it is very interesting to show that our broad emission can be reproduced with the peak position at 671 nm in the double-layered $PEA_2PbI_4/PEA_2SnI_4$ interfaces prepared by our hand–finger pressing method (Fig. 3b), very similar to the broad emission observed by mixing two precursor solutions. Because our hand–finger pressing method can only form a low density of double-layered $PEA_2PbI_4/PEA_2SnI_4$ heterostructures, the broad emission becomes an appreciable shoulder on the intrinsic emission peaked at 638 nm from the $PEA_2SnI_4$ perovskite.

Clearly, the similar broad emission observed from our hand–finger pressing method as compared to the precursor-solution mixing method provides an unambiguous evidence to confirm that the broad emission indeed represents the CTEs formed at the interfaces between $PEA_2PbI_4$ and $PEA_2SnI_4$ domains in 2D perovskite heterostructures.

**Dynamic analysis of CTE formation.** To explore the dynamic behaviors of the CTEs, we performed pump–probe TA studies on the heterostructured 2D perovskite $[(PEA_2PbI_4)_{0.95}:(PEA_2SnI_4)_{0.05}]$ film (Fig. 4a). The TA characteristics show an interesting phenomenon where a broad signal is observed between 600 and 800 nm (bottom panel in Fig. 4a). Obviously, the broad TA signal is consistent with the broad emission from CTEs in the heterostructured 2D perovskite film. Contrarily, the pristine $PEA_2PbI_4$ film does not show any broad TA signal between 600 and 800 nm, only leading to a photobleaching signal peaked at 514 nm related to intrinsic excitons, as shown in Fig. 4b. Clearly, the broad TA signal provides the dynamic information of CTEs formed in 2D perovskite heterostructures. Figure 4c summarizes the dynamic characteristics of CTEs with the broad TA signal between 600 and 800 nm occurring at the $[PEA_2PbI_4:PEA_2SnI_4]$ interfaces in 2D perovskite heterostructures $[(PEA_2PbI_4)_{0.95}:(PEA_2SnI_4)_{0.05}]$. The CTEs shown as the broad TA signal between 600 and 800 nm are quickly formed in 3 ps and slowly generate a light emission after 230 ps. Interestingly, the broad photoinduced bleaching TA signal extends to a few nanoseconds without any appreciable decay. This indicates that the CTEs are essentially metastable states in 2D perovskite heterostructures. Furthermore, the TA characteristics show a spectral shoulder at 537.5 nm in the heterostructured 2D perovskite film, slightly below the emission from intrinsic excitons (peaked at 513.8 nm) in the $PEA_2PbI_4$ domains. This spectral shoulder (537.5 nm) is formed in 1 ps, quickly demonstrates a light emission after 2.2 ps, and decays within nanosecond time window. By considering the lower energy (51 meV below the band-to-band transition) and photobleaching developed after the band-to-band transition, we assigned the TA spectral shoulder (peaked at 537.5 nm) to surface-trapped excitons. Because the CTEs are developed into a bleaching signal after 230 ps, the surface-trapped excitons, which are quickly shown as a bleaching signal within 2.2 ps, can have an opportunity to contribute to the formation of CTEs. Furthermore, the surface-trapped excitons have a limited light emission in our 2D perovskite heterostructured film. It is

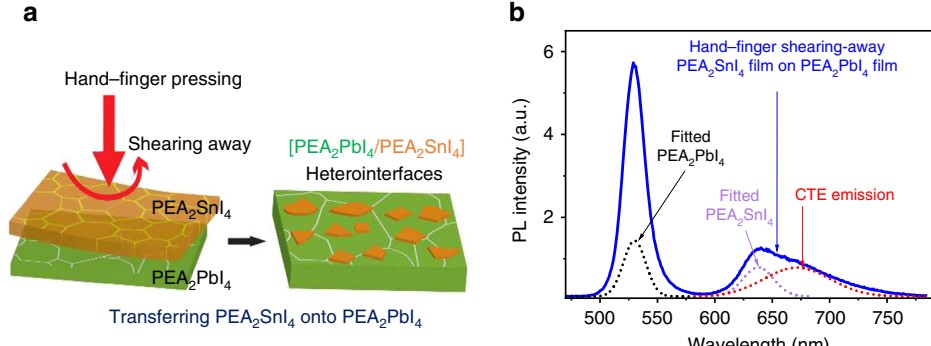

**Fig. 3 Hand–finger pressing method to confirm CTE formation in 2D perovskite heterostructures. a** Schematic diagram illustrates that $PEA_2SnI_4$ film is shearing away on $PEA_2PbI_4$ film surface to prepare double-layered $[PEA_2PbI_4/PEA_2SnI_4]$ heterointerfaces. **b** PL spectrum was measured for double-layered $[PEA_2PbI_4/PEA_2SnI_4]$ heterointerfaces. The broad emission is shown as an extended spectral tail in addition to intrinsic emission (530 and 637.5 nm) from $PEA_2PbI_4$ and $PEA_2SnI_4$ components. The dashed lines are shown for three spectral components from $PEA_2PbI_4$ component, $PEA_2SnI_4$ component, and $[PEA_2PbI_4/PEA_2SnI_4]$ heterointerfaces.

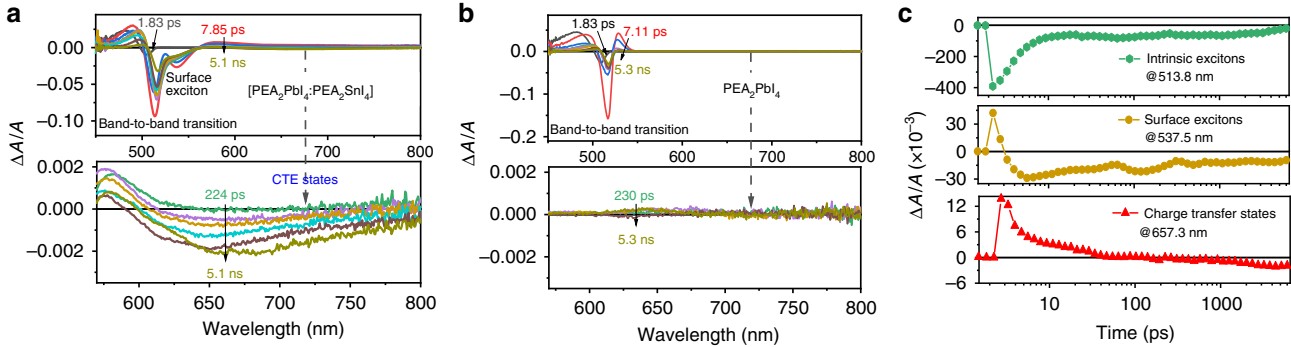

**Fig. 4 Transient absorption (TA) characteristics for 2D perovskite heterostructures and PEA$_2$PbI$_4$. a** TA spectra for 2D perovskite heterostructured [(PEA$_2$PbI$_4$)$_{0.95}$:(PEA$_2$SnI$_4$)$_{0.05}$] film. **b** TA spectra for pristine PEA$_2$PbI$_4$ film. **c** TA dynamics plotted for band-to-band transition (513.8 nm), surface excitons (537.5 nm), and broad emission (657.3 nm) in 2D perovskite heterostructures.

shown in Supplementary Fig. 8 that the PL of pure PEA$_2$PbI$_4$ film contains a weak tail located around 540 nm, slightly below the emission (peaked at 520 nm) from intrinsic excitons with decreased film thickness (~20 nm), leading to an asymmetric spectrum. This weak PL tail at 540 nm coincides with the surface-trapped excitons at 537.5 nm shown in TA (Fig. 4a). With increasing the film thickness to ~150 nm, this weak tail becomes negligible, making the entire PL spectrum symmetric. We believe this weak tail on PL spectrum at thinner film provides a further indication to support the surface-trapped excitons. As a comparison, the intrinsic excitons shown as the band-to-band transition peaked at 513.8 nm are quickly developed into a light emission within ~1 ps. Therefore, the dynamics of CTEs include the following three processes. First, both intrinsic and surface-trapped excitons experience a quick charge transfer, leading to the formation of CTEs at the heterointerfaces between host and guest perovskite structures within 3 ps. Second, the CTEs formed at heterostructures gradually develop a broad light emission, showing a TA bleaching signal after 230 ps. Third, the light emission of CTEs exhibits an extended lifetime (~0.5 μs), indicating that the CTEs are metastable states in 2D perovskite heterostructured film. To further verify the formation of CTEs, we have also measured TA for the double-layered heterointerfaces prepared by our hand–finger pressing method (Supplementary Fig. 9). The double-layered heterointerfaces demonstrate a broad TA signal between 640 and 800 nm, in addition to the band-to-band transitions (peaked at 520 and 590 nm) in Pb and Sn perovskites. This broad TA signal observed from the double-layered heterointerfaces prepared by our hand–finger pressing method provides a further evidence to confirm the formation of CTEs.

**Heterostructured 2D perovskite photodetectors and LEDs.** Now we use below-bandgap photoexcitation to show that the CTEs are vital states formed in 2D perovskite heterostructures. Figure 5a illustrates the photocurrents for pristine (PEA$_2$PbI$_4$, PEA$_2$SnI$_4$) and heterostructured [(PEA$_2$PbI$_4$)$_{0.95}$:(PEA$_2$SnI$_4$)$_{0.05}$] films with the device architecture of ITO/poly (3,4-ethylene-dioxythiophene) polystyrene sulfonate (PEDOT:PSS)/perovskite/phenyl-C$_{61}$-butyric acid methyl (PC$_{61}$BM)/PEI/Ag under the photoexcitation of 640 nm. We should note that the 2D heterostructured [(PEA$_2$PbI$_4$)$_{0.95}$:(PEA$_2$SnI$_4$)$_{0.05}$] and the pristine PEA$_2$PbI$_4$ films have the same absorbance at the wavelength of 640 nm (selected as the below-bandgap photoexcitation) (Fig. 5b). Interestingly, the heterostructured [(PEA$_2$PbI$_4$)$_{0.95}$:(PEA$_2$SnI$_4$)$_{0.05}$] film demonstrates a much enlarged ON–OFF ratio of 532 on the photocurrent when the 640 nm photoexcitation is applied and removed periodically. This below-bandgap

photoexcitation-induced photocurrent directly indicates that the charge-transfer states are vital states formed in 2D perovskite heterostructures. In contrast, both pristine PEA$_2$PbI$_4$ and PEA$_2$SnI$_4$ films show significantly reduced ON–OFF ratios of 29.7 and 3.91 on photocurrent upon repeatedly applying and removing the 640 nm photoexcitation. Furthermore, we observed that the CTEs can be generated by electrically injected charge carriers in 2D perovskite heterostructures. Figure 5c shows the electroluminescence (EL) spectrum from the heterostructured 2D perovskite [(PEA$_2$PbI$_4$)$_{0.95}$:(PEA$_2$SnI$_4$)$_{0.05}$] film based on the device design of ITO/PEDOT:PSS/poly(9-vinylcarbazole) (PVK)/perovskite/2,2′,2″-(1,3,5-benzinetriyl)-tris(1-phenyl-1-*H*-benzimidazole) (TPBi)/lithium fluoride (LiF)/Ag. The EL spectrum combines the emission from both intrinsic excitons (peaked at 517 nm) from the host PEA$_2$PbI$_4$ component and the CTEs (peaked at 669 nm) formed at the interfaces between PEA$_2$PbI$_4$ and PEA$_2$SnI$_4$ components in 2D perovskite heterostructures [(PEA$_2$PbI$_4$)$_{0.95}$:(PEA$_2$SnI$_4$)$_{0.005}$]. The EL–voltage–current characteristics shown in Fig. 5d indicate the typical LED behaviors from CTEs under electrical injection. It should be pointed out that the injection current is quite low: (<1 mA/cm$^2$) due to the lower conductance caused by insulating organic long-chain ligands in 2D perovskite film even when the applied bias reaches 13 V. Clearly, the CTEs allow the generation of broad EL in 2D perovskite heterostructures under electrical injection.

## Discussion

In summary, we found that the CTEs can be formed at the interfaces between PEA$_2$PbI$_4$ and PEA$_2$SnI$_4$ components in 2D perovskite heterostructures [(PEA$_2$PbI$_4$)$_{1-x}$:(PEA$_2$SnI$_4$)$_x$] prepared by chemically mixing two precursor (PEA$_2$PbI$_4$ and PEA$_2$SnI$_4$) solutions. The PEA$_2$PbI$_4$ and PEA$_2$SnI$_4$ serve as host and guest components in the heterostructured 2D perovskite [(PEA$_2$PbI$_4$)$_{0.999}$:(PEA$_2$SnI$_4$)$_{0.001}$] films. A broad emission peaked at 669 nm with spectral range between 575 and 800 nm was observed with the intensity-dependence slope of 1.62 and the prolonged lifetime (~0.5 μs) in such 2D perovskite heterostructures. Particularly, when the host and guest components are alternatively switched to shift the bandgap largely from 2.33 eV in the [(PEA$_2$PbI$_4$)$_{0.999}$:(PEA$_2$SnI$_4$)$_{0.001}$] to 1.95 eV in the [(PEA$_2$SnI$_4$)$_{0.999}$:(PEA$_2$PbI$_4$)$_{0.001}$], the broad emission spectrum is only slightly changed from 669 nm (1.85 eV) to 697 nm (1.78 eV) in peak position, leading to similar broad emission between PEA$_2$PbI$_4$ and PEA$_2$SnI$_4$ based heterostructures. This leads to the hypothesis that the CTEs can be conveniently established in 2D perovskite heterostructures. This hypothesis was verified by the unique phenomenon: a similar broad emission can be reproduced

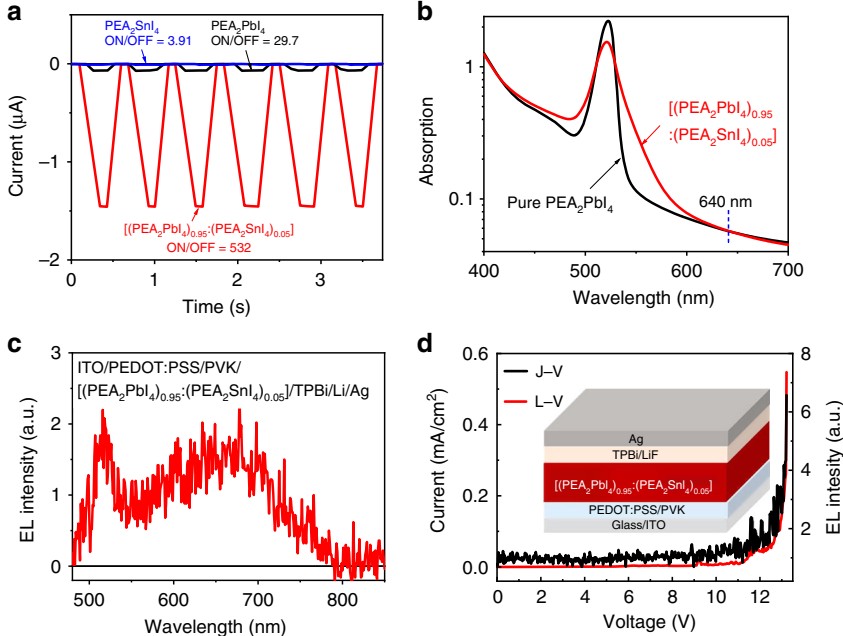

**Fig. 5 Device performance of photodetectors and LEDs for 2D perovskite heterostructures. a** The ON–OFF behaviors of heterostructured 2D perovskite $[(PEA_2PbI_4)_{0.95}:(PEA_2SnI_4)_{0.05}]$ photodetector (red) as compared to pure $PEA_2PbI_4$ (black) and $PEA_2SnI_4$ (blue) devices when exciting at 640 nm (1000 mW/cm²). **b** Absorption spectra of two films. **c** EL spectrum containing broad CTE emission and host emission. **d** Current density–voltage–EL curve of doped device. The device structure is shown as the inset.

when the $PEA_2SnI_4$ film was shearing away onto the surface of the $PEA_2PbI_4$ film by hand–finger pressing method. The pump–probe TA studies show that the CTEs are formed within 3 ps between $PEA_2PbI_4$ and $PEA_2SnI_4$ components and slowly developed into a photobleaching signal after 230 ps. Particularly, the photobleaching signal from CTEs extends a few nanoseconds without appreciable decay. Essentially, the CTEs become metastable states in 2D perovskite heterostructures. The photocurrent generated by below-gap photoexcitation shows that the charge-transfer states are vital states formed at the interfaces between $PEA_2PbI_4$ and $PEA_2SnI_4$ components in heterostructured 2D perovskite films. The EL indicates that the CTEs can be formed by electrically injected electrons and holes to generate a broad light emission in perovskite LEDs. Therefore, the CTEs present new opportunities to develop advanced optoelectronic properties based on heterostructural design in 2D perovskites.

## Methods

**Thin-film preparation**. Lead iodide (PbI₂), tin iodide (SnI₂), PVK, TPBi, LiF, chlorobenzene, isopropyl alcohol (IPA), and dimethyl sulfoxide (DMSO) were purchased from Sigma-Aldrich. Phenethylammonium iodide (PEAI) was purchased from Greatcell Solar. PC₆₁BM was purchased from 1-Materials. PEDOT: PSS was purchased from Clevios. All chemicals are used as received without further purification. The precursor solution of $PEA_2PbI_4$ was prepared by mixing PEAI (0.8 M) and PbI₂ (0.4 M) in DMSO (1 mL). The precursor solution of $PEA_2SnI_4$ was made in the same way by replacing PbI₂ with SnI₂. Heterostructures of $[(PEA_2PbI_4)_{1-x}:(PEA_2SnI_4)_x]$ were designed by mixing two precursor ($PEA_2PbI_4$ and $PEA_2SnI_4$) solutions with alternatively selected high and low volume ratios. All the perovskite films were fabricated by spin coating the precursor solutions on glass slides (1.5 × 1.5 cm²) at 3000 r.p.m. for 30 s, and then thermally annealed at 100 °C for 10 min.

**Fabrications of LED and photodetector**. The patterned ITO substrates were first cleaned by ultrasonic treatments with detergent, deionized water, acetone, and isopropanol, respectively, for 20 min in each step and dried with nitrogen gas, and then followed by ultraviolet treatment for 30 min. After that, the PEDOT:PSS hole-transport layer of 40 nm was spin coated on the precleaned ITO substrates at 4000 r.p.m. for 60 s, and then annealed at 150 °C for 30 min. For LED devices, a thin

layer of PVK (16 mg/ml dissolved in chlorobenzene) was spin coated on the PEDOT:PSS layers with a speed of 5000 r.p.m. for 60 s, and treated with the thermal annealing at 120 °C for 20 min. The 2D perovskite precursor solutions were spin coated as described above. The TPBi was vacuum deposited with the thickness of 50 nm. Two nanometers of LiF and 90 nm silver were evaporated to prepare the perovskite LEDs with the architecture of ITO/PEDOT:PSS/PVK/perovskite/TPBi/LiF/Ag. For the fabrication of photodetector devices, the PC₆₁BM (25 mg/ml dissolved in chlorobenzene) was spin coated at 2000 r.p.m. for 50 s, without thermal annealing, serving as an electron-transport layer, and then an interface modification layer of PEI (0.5 mg/ml, IPA) was spin coated at 5000 r.p.m for 60 s, followed by the evaporation of silver electrode of 90 nm. This leads to the photodetector device architecture of ITO/PEDOT:PSS/perovskite/PC₆₁BM/PEI/Ag. The active area for both LED and photodetector devices is 0.06 cm².

**Mechanically preparing heterostructures**. The double-layer 2D perovskite heterostructures were prepared by using hand–finger pressing method to avoid lattice distortion and doping effect. Specifically, the $PEA_2SnI_4$ film coated on glass substrate was held on the $PEA_2PbI_4$ film with the face-to-face contact by using hand–finger pressing method. Then, the $PEA_2SnI_4$ film was shearing on the surface of $PEA_2PbI_4$ film. This leaves the $PEA_2SnI_4$ film islands on the $PEA_2PbI_4$ film, leading to double-layer heterointerfaces ($PEA_2SnI_4$/$PEA_2PbI_4$). The $PEA_2SnI_4$ film islands can be visible as pale-yellow color on the $PEA_2PbI_4$ film. Therefore, our hand–finger pressing method can prepare the heterostructures without lattice distortion and doping effects to confirm the charge-transfer excitons responsible for broad emission observed in 2D perovskite heterostructures.

**Characterizations and measurements**. The steady-state and time-resolved PL characteristics were measured with Flouro Log III spectrometer. The absorption spectrum was measured with UV3600 (Shimadzu, Japan). TA spectra were collected by using a Helios Fire spectrometer (Ultrafast Systems LLC). The pump beam (346 nm, 5 μW) was generated through a harmonic generator (Ultrafast Systems LLC, third harmonic) pumped by a Pharos laser (Light Conversion, 1 kHz, 1030 nm, 290 fs). The probe beam was selected from the broad light emission generated by CaF₂ crystal excited by 1030 nm fs primary beam.

## Data availability

The data that support the findings of this study are available from the corresponding author on reasonable request.

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

## Acknowledgements

B.H. acknowledges the funding support from the National Science Foundation (NSF-1911659) in the United States. This research was partially conducted at the Center for Nanophase Materials Sciences based on user projects (CNMS2016-279, CNMS2016-R45), which is sponsored by Oak Ridge National Laboratory by the Division of Scientific User Facilities, US Department of Energy. The author X.X.Z. (visiting Ph.D. student) acknowledges the support from the National Natural Science Foundation of China (Grant Nos. 61634001, U1601651, 61475051, and 61604010).

## Author contributions

B.H. directed the research project and supervised the experimental design. J.Z. conceived and performed the experiment studies and analyzed the data. X.X.Z. and M.S.W. contributed to the fabrication of LEDs. The manuscript was prepared by both J.Z. and B.H.

## Competing interests

The authors declare no competing interests.
