## [Peer Review File · Nature Communications]

Reviewers' Comments:

Reviewer #1:

Remarks to the Author:

This work by Zhang et al presents an interesting study, where they found signature of CT excitation at 2D perovskite hetero-interfaces. Specifically, they observed a broadened low energy feature in the PL spectrum when making this heterostructure, and it formed quickly and decay slowly. It is also interesting that they can reproduce such feature by mechanical contacts. Finally, the authors have probed the CT signatures by photo current and EL in a device configuration. While this study is quite new and interesting, I see some open questions from this work. I would suggest to address those before being considered for Nat Comm.

1. The PL signature for the CT feature identified by the authors worth further investigation. For instance, how would one distinguish the exciton feature from interface trap feature? The authors investigated the power dependence and attributed it to "non-germinate" recombination. However, many processes could follow such recombination. Also, the authors did not comment on the long-decay lifetime, which is usually not so long in CTE in organic systems.
2. Structure characterization is highly recommended for this study, the claimed hetero-structures in this work have only been probed by optical spectroscopy tools. I would suggest to at least use XRD or AFM to validate their claim.
3. Same comment applies for Fig. 3, the surface feature should be probed by AFM if possible.
4. Fig. 5d is not well discussed. I wonder why such a high voltage is needed to turn on the device, and the injection current is very low, $<1\text{mA/cm}^2$ at 12 volt. A common LED or solar cell, should show much larger current at 12 volt.

Reviewer #2:

Remarks to the Author:

In this paper, authors attempt to establish charge-transfer excitons in 2D perovskite heterointerfaces, which can be formed by either mixing lead and tin precursor solutions or using hand-finger pressing method. Despite explosive interest in perovskite optoelectronic devices, researches on charge-transfer excitons in 2D perovskites are rarely reported. The aims of this work are therefore important. Although there are some inconclusive arguments within the manuscript, authors try their best to figure out everything behind the charge-transfer excitons. Overall, this is good work.

1. This work mainly focuses on charge-transfer excitons of 2D perovskite heterointerfaces, but some basic characterizations, such as SEM, XRD, and AFM seem to be required to understand.
2. In general, charge-transfer excitons have a shorter PL lifetime than those of intrinsic excitons. However, we note that the PL lifetime for broad emission (peaked at 669 nm) in 2D perovskite heterostructures are the longest among the three samples in Figure 2c. Please double check the PL lifetime and explain this discrepancy.
3. In addition to the intrinsic excitons, the surface-trapped excitons are also observed in transient absorption spectra (Figure 4a). The presence of surface-trapped excitons may considerably vary the charge-carrier dynamics.
4. As mentioned above, the surface-trapped excitons do exist close to surface regions, so characterizing the electronic structure at the contact interface between two types of perovskites is of critical importance to understand this phenomenon.
5. To corroborate the origin of broad emission in the PEA₂PbI₄/PEA₂SnI₄ heterostructure, the transient absorption for samples formed by using the hand-finger pressing method will help to provide strong pieces of evidence.
6. I am very happy to see the electroluminescence data showing red-shifted broad emission, which is similar to the PL results. In contrast to LEDs, photodetectors may be more suitable for evaluating their practical application.
7. Minor revisions:

- a) There is no space between the number and units (Figure 2 captions), please carefully check similar issues throughout the manuscript.
- b) The charge-transfer states at the 2D perovskite/organic interfaces have been discussed in the literature. Please add some discussion in the introduction section.

Reviewer #3:

Remarks to the Author:

The manuscript by Hu and co-authors has presented an interesting study on the formation of charge-transfer excitons in quasi-2D perovskite heterostructures. To demonstrate the possibility of CT exciton formation, they used mixed Pb-Sn quasi-2D perovskite as a model system for experimental investigations. The spectrally-broad photoemission from PEA₂PbI₄:PEA₂SnI₄ has been explained in terms of CT excitons, and has been supported by an interesting finger pressing experiment. The CT exciton formation process is studied using transient optical experiments. Overall, the novelty and importance of the paper clearly fall within the scope of Nature Communications. Before recommending publication, the authors are advised to improve their paper by considering the following relatively minor points.

1. In the introductory paragraph (page 3), the authors state that: "undoubtedly, the excitons formed within intrinsic perovskite structures are solely responsible for developing high-performance light emitting properties". The meaning of this statement is rather unclear. Are excitons formed solely responsible for light emitting properties? Apart from recombining radiatively (to emit light) or non-radiatively (to produce heat), excitons may also diffuse and dissociate in a LED device. The authors should consider rewriting this sentence to make it clearer. Also, tuning light emitting properties in hybrid halide perovskite is not experimentally complex. An example of this is the very simple emission color tuning using a mixed halide approach.
2. The relatively broad luminescent spectra and the transient optical experiments of the excitations show interesting similarities to what have been reported earlier in Ref 19, where heterostructures formed by (quasi-)2D perovskite, 3D perovskite and polymer were investigated. Ref 19 showed that charge-separated states in the heterostructures were formed within 1 ps before they radiatively recombined more slowly through a bi-molecular process. It was unclear whether the heterostructures were energetically aligned (e.g., type I or type II), but a similar energetic disorder was certainly present. It may be useful for the authors to discuss/comment on these relevant results in relation to the experimental findings of the current paper.
3. To further demonstrate the presence of CTE, the authors measured the photocurrents generated by the perovskite films under sub-bandgap photoexcitation. While this is certainly a nice experiment to do, the reviewer is not 100% convinced that the wavelength of photoexcitation can be considered truly sub-bandgap. The reason for this argument is that the apparent optical bandgap of the PEA₂PbI₄(minority):PEA₂SnI₄(majority) as reported by the authors is 1.95 eV. I presume PEA₂SnI₄ should have very similar (if not the same) bandgap. 1.95 eV is equivalent to a photon wavelength of 635 nm. The authors used a light source with a central wavelength of 640 nm, only 5 nm below the bandgap photon, to provide the photoexcitation. This is very close to the absorption edge of the material. It means that it would not be surprising to observe photocurrent for the semiconductor, as the absorption coefficient at this spectral region so close to the apparent optical bandgap is still not sufficiently close to zero. The type and spectral bandwidth of the light source used are also not specified. The authors should come up with an improved experiment with considerably lower photon energies (much lower than $E_g - kT$). If doing such new experiment is not possible, the authors should comment on the potential weakness of the relevant conclusion drawn from the current experimental setup.

Responses to Reviewer #1

Review comment

This work by Zhang et al presents an interesting study, where they found signature of CT excitation at 2D perovskite hetero-interfaces. Specifically, they observed a broadened low energy feature in the PL spectrum when making this heterostructure, and it formed quickly and decay slowly. It is also interesting that they can reproduce such feature by mechanical contacts. Finally, the authors have probed the CT signatures by photo current and EL in a device configuration. While this study is quite new and interesting, I see some open questions from this work. I would suggest to address those before being considered for Nat Comm.

Author response

We thank the referee for the recommendation and suggestion. In the revised manuscript, we have added systematic experiments including structure characterizations (XRD, AFM and SEM) and additional TA data of mechanical contacts, to further explore the mechanism of CTEs. Therefore, a clear physics picture of how CTEs are formed is described as follows. 1. The excitons are formed in pure 2D perovskite nanoplates (Pb based or Sn based host) upon photoexcitation; 2. The excitons formed nearby the surfaces of nanoplates experience a quick charge-transfer process at hetero-interfaces between host and guest perovskite structures within less than 3 ps. 3. Upon charge-transfer process, the electron-hole pairs are formed as charge-transfer excitons at hetero-interfaces, leading to broad light emission with extended lifetime ($\sim 0.5 \mu\text{s}$). We have added these further explanations into the revised manuscript (line 23 on page 12 to line 6 on page 13).

Review comment 1

1. The PL signature for the CT feature identified by the authors worth further investigation. For instance, how would one distinguish the exciton feature from interface trap feature? The authors investigated the power dependence and attributed it to “non-germinate” recombination. However, many processes could follow such recombination. Also, the authors did not comment on the long-decay lifetime, which is usually not so long in CTE in organic systems.

Author response 1

We thank the referee for the questions/comments. In general, the excitons and trapped excitons can be normally distinguished by two parameters: energy and lifetime. In our TA studies (Fig. 4 in the manuscript), we observed a strong bleaching signal at 513.8 nm, very quickly appeared within sub-picoseconds, which can be assigned to band-to-band transition that coincides with

exciton emission in wavelength. Therefore, this strong bleaching signal at 513.8 nm presents the dynamics of excitons in 2D perovskite heterostructured [(PEA₂PbI₄)_{0.95}:(PEA₂SnI₄)_{0.05}] film.

Furthermore, we observed a small shoulder peak at 537.5 nm in TA spectrum, appeared within 1 ps and then developed a bleaching signal within 2.2 ps after the band-to-band bleaching is quickly developed. Obviously, this shoulder peak (at 537.5 nm) lies 51 meV below the excitons (shown as the band-to-band-transition at 513.8 nm). By considering the lower energy (51 meV below the band-to-band transition) and bleaching developed after the band-to-band transition, we assigned the shoulder peak (at 537.5 nm) to surface-trapped excitons in 2D perovskite heterostructured [(PEA₂PbI₄)_{0.95}:(PEA₂SnI₄)_{0.05}] film. We have added these further discussions into the revised manuscript (line 9 to line 14 on page 12).

In general, the lifetime of CTEs is mainly determined by the electron-hole recombination rate governed by Coulomb attractive force. As a result, CTEs often exhibit an extended lifetime due to longer electron-hole separation distance as compared to intrinsic excitons in semiconducting materials. We should note that CTEs can still demonstrate various lifetimes when the Coulomb attraction between electron and hole located on different energetic structures is changed by local dielectric backgrounds, widely observed in organic-organic¹⁻³, organic-inorganic^{4,5}, inorganic-inorganic mixtures^{6,7}. We have added these further discussions into the revised manuscript (line 11 to line 18 on page 8).

We agree with the referee that many processes can follow the recombination with the power dependence slope of 2. For example, two-photon absorption-induced up-conversion photoluminescence also shows the power dependence slope of 2. However, the non-germinate recombination can give the power dependence slope of 2 in the situation where electrons and holes have the same numbers. Specifically, when electrons and holes are paired to form electron-hole pairs, the pairing probability p is proportional to the product between the number (n) of electrons and the number (m) of holes: $p \propto m \cdot n$. If $n=m$, then $p \propto n^2$. We should note that germinate process dominates the radiative recombination of the intrinsic excitons in host/guest 2D perovskites due to large exciton binding energy, which leads to the power dependence slope of ~ 1 . In our heterostructured [(PEA₂PbI₄)_{0.95}:(PEA₂SnI₄)_{0.05}] film, the power dependence slope of 1.62 indicates the non-germinate recombination (with which the slope is 2), implying that the broad emission carries the signature of electron-hole recombination. It provides an evidence

of charge-transfer excitons in the 2D perovskite heterostructures [(PEA₂PbI₄)_{0.999}:(PEA₂SnI₄)_{0.001}]. We have added these further discussions into the revised manuscript (line 22 on page 7 to line 7 on page 8).

Review comment 2

2. Structure characterization is highly recommended for this study, the claimed heterostructures in this work have only been probed by optical spectroscopy tools. I would suggest to at least use XRD or AFM to validate their claim.

Author response 2

We thank the referee for the suggestion. We have added XRD, AFM and SEM results in the supporting information of revised manuscript (see supplementary Fig. S1, S2, and S3). According to XRD, we can see that all the 2D perovskite films show excellent crystallinity with extremely narrow dominant peak at (002) direction. Furthermore, the XRD peak of PEA₂SnI₄ film is slightly redshifted as compared with the PEA₂PbI₄ film due to smaller crystal size. By increasing the ratio of PEA₂SnI₄ in 2D perovskite heterostructured film [(PEA₂PbI₄)_{0.95}:(PEA₂SnI₄)_{0.05}], the XRD peak is redshifted towards that of PEA₂SnI₄ film. Moreover, the XRD intensity is also decreased towards that of PEA₂SnI₄ film. This provides an evidence that in the mixed heterostructured film, both PEA₂PbI₄ and PEA₂SnI₄ perovskites are formed. Both SEM and AFM results show that all the films are of high quality. We have added these further discussions into the revised manuscript (line 18 on page 5 to line 5 on page 6).

Supplementary Fig. S1. X-Ray Diffraction (XRD) patterns obtained from PEA_2PbI_4 , $[(\text{PEA}_2\text{PbI}_4)_{0.99}:(\text{PEA}_2\text{SnI}_4)_{0.01}]$, $[(\text{PEA}_2\text{PbI}_4)_{0.95}:(\text{PEA}_2\text{SnI}_4)_{0.05}]$, $[(\text{PEA}_2\text{PbI}_4)_{0.05}:(\text{PEA}_2\text{SnI}_4)_{0.95}]$ and PEA_2SnI_4 films. All films show an in-plane orientation of the $[\text{PbX}_6]^{4-}$ sheets along substrate surface.

Supplementary Fig. S2. AFM results of three different perovskite films: (a) PEA_2PbI_4 . (b) $(\text{PEA}_2\text{PbI}_4)_{0.95}:(\text{PEA}_2\text{SnI}_4)_{0.05}$. (c) PEA_2SnI_4 .

Supplementary Fig. S3. SEM results of three different perovskite films: (a) PEA_2PbI_4 . (b) $(\text{PEA}_2\text{PbI}_4)_{0.95}:(\text{PEA}_2\text{SnI}_4)_{0.05}$. (c) PEA_2SnI_4 .

Review comment 3

3. Same comment applies for Fig. 3, the surface feature should be probed by AFM if possible.

Author response 3

We thank the referee for the suggestion. We encountered a technical difficulty to take AFM measurement of our double-layered heterostructured film [PEA₂PbI₄/PEA₂SnI₄] prepared by our hand-finger pressing method, due to surface roughness. But we were successful to use microscope photos under white light illumination to reveal the heterointerfaces in the double-layered film, shown in supplementary Fig. S7a. We can clearly see the red islands from Sn perovskite (PEA₂SnI₄) formed on the green-yellow background from Pb perovskite (PEA₂PbI₄) film, confirming the double-layered heterointerfaces prepared by our hand-finger pressing method, shown in supplementary Fig. S7a. The supplementary Fig. S7b shows the Pb perovskite-only (PEA₂PbI₄) film under white light illumination, as shown as green-yellow background. The slightly non-uniform color on Pb perovskite-only film is caused by the uneven surface through white light illumination. We have added the microscope photos into the revised manuscript (see supplementary Fig. S7 below).

Supplementary Fig. S7. Microscope photos under white light illumination. (a) Double-layered [PEA₂PbI₄/PEA₂SnI₄] film prepared by hand-finger pressing method. (b) pure PEA₂PbI₄. The image size is 54*43 μm^2 .

Review comment 4

4. Fig. 5d is not well discussed. I wonder why such a high voltage is needed to turn on the device, and the injection current is very low, <1mA/cm² at 12 volt. A common LED or solar cell, should show much larger current at 12 volt.

Author response 4

We thank the referee for the suggestion. We like to mention that, although the applied bias is high (13 V), the injection current is quite low: (< 1 mA/cm²). Here, the injection current density is limited by the lower conductance in 2D perovskite films due to the existence of insulating organic long-chain ligands. The low injection currents are also be observed in other 2D perovskite devices^{8,9}. We have discussed this issue in the revised manuscript (line 19 to line 22 on page 14).

Responses to Reviewer #2

Review comment

In this paper, authors attempt to establish charge-transfer excitons in 2D perovskite heterointerfaces, which can be formed by either mixing lead and tin precursor solutions or using hand-finger pressing method. Despite explosive interest in perovskite optoelectronic devices, researches on charge-transfer excitons in 2D perovskites are rarely reported. The aims of this work are therefore important. Although there are some inconclusive arguments within the manuscript, authors try their best to figure out everything behind the charge-transfer excitons. Overall, this is good work.

Author response

We thank the referee for this encouraging comment.

Review comment 1

1. This work mainly focuses on charge-transfer excitons of 2D perovskite heterointerfaces, but some basic characterizations, such as SEM, XRD, and AFM seem to be required to understand.

Author response 1

We thank the referee for the suggestions. We have added XRD, AFM and SEM results in the supporting information in this revised manuscript (see supplementary Fig. S1, S2 and S3 below). According to the XRD, we can see that all the 2D perovskite films show excellent crystallinity with extremely narrow dominant peak at (002) direction. Specifically, the XRD peak of the PEA_2SnI_4 film is slightly redshifted as compared with the PEA_2PbI_4 film due to smaller crystalline size. By increasing the ratio of PEA_2SnI_4 in the mixed heterostructured $[(\text{PEA}_2\text{PbI}_4)_{0.999}:(\text{PEA}_2\text{SnI}_4)_{0.001}]$ film, the XRD peak is redshifted towards that of PEA_2SnI_4 film. Moreover, the XRD intensity is also decreased towards that of PEA_2SnI_4 film. This provides an evidence that in the mixed heterostructured $[(\text{PEA}_2\text{PbI}_4)_{0.999}:(\text{PEA}_2\text{SnI}_4)_{0.001}]$ film, both PEA_2PbI_4 and PEA_2SnI_4 perovskites are formed. SEM and AFM results show that all the films are of high quality. We have added these further discussions into the revised manuscript (line 18 on page 5 to line 5 on page 6).

Supplementary Fig. S1. X-Ray Diffraction (XRD) patterns obtained from PEA_2PbI_4 , $[(\text{PEA}_2\text{PbI}_4)_{0.99}:(\text{PEA}_2\text{SnI}_4)_{0.01}]$, $[(\text{PEA}_2\text{PbI}_4)_{0.95}:(\text{PEA}_2\text{SnI}_4)_{0.05}]$, $[(\text{PEA}_2\text{PbI}_4)_{0.05}:(\text{PEA}_2\text{SnI}_4)_{0.95}]$ and PEA_2SnI_4 films. All films show an in-plane orientation of the $[\text{PbX}_6]^{4-}$ sheets along substrate surface.

Supplementary Fig. S2. AFM results of three different perovskite films: (a) PEA_2PbI_4 . (b) $(\text{PEA}_2\text{PbI}_4)_{0.95}:(\text{PEA}_2\text{SnI}_4)_{0.05}$. (c) PEA_2SnI_4 .

Supplementary Fig. S3. SEM results of three different perovskite films: (a) PEA_2PbI_4 . (b) $(\text{PEA}_2\text{PbI}_4)_{0.95}:(\text{PEA}_2\text{SnI}_4)_{0.05}$. (c) PEA_2SnI_4 .

Review comment 2

2. In general, charge-transfer excitons have a shorter PL lifetime than those of intrinsic excitons. However, we note that the PL lifetime for broad emission (peaked at 669 nm) in 2D perovskite heterostructures are the longest among the three samples in Figure 2c. Please double check the PL lifetime and explain this discrepancy.

Author response 2

We thank the referee for the question/suggestion. Before submission, we have experimentally double-checked the long PL lifetime of charge-transfer excitons multiple times by comparing the heterostructured and pristine perovskite films. In general, the lifetime of CTEs is mainly determined by the electron-hole recombination rate governed by Coulomb attractive force. As a result, CTEs often exhibit an extended lifetime due to longer electron-hole separation distance as compared to intrinsic excitons in semiconducting materials. We should note that CTEs can still demonstrate various lifetimes when the Coulomb attraction between electron and hole located on different energetic structures is changed by local dielectric backgrounds, widely observed in organic-organic¹⁻³, organic-inorganic^{4,5}, inorganic-inorganic mixtures^{6,7}. We have added this explanation into the revised manuscript (line 11 to line 18 on page 8).

Review comment 3

3. In addition to the intrinsic excitons, the surface-trapped excitons are also observed in transient absorption spectra (Figure 4a). The presence of surface-trapped excitons may considerably vary the charge-carrier dynamics.

Author response 3

We thank the referee for the comment. In our study, the surface-trapped excitons were derived from the spectral shoulder (peaked at 537.5 nm) with 51 meV slightly below the band-to-band bleaching (513.8 nm) in our transient absorption spectrum. This shoulder is appeared within 1 ps and then developed a bleaching signal within 2.2 ps after the band-to-band bleaching is quickly developed. By considering the lower energy (51 meV below the band-to-band transition) and bleaching developed after the band-to-band transition, we assigned the shoulder peak (at 537.5 nm) to surface-trapped excitons in 2D perovskite heterostructured [(PEA₂PbI₄)_{0.95}:(PEA₂SnI₄)_{0.05}] film. In our work, the surface-trapped excitons are appeared within 1 ps while the CTEs are appeared within 3 ps shown in our transient absorption spectrum. Then, the CTEs are developed into a bleaching signal after 230 ps while the surface-trapped excitons are quickly shown a bleaching signal within 2.2 ps. For such dynamics, we can see that the surface-trapped excitons can develop into CTEs. The power dependence slope of 1.62 (Fig. 2b in our revised manuscript) suggests that there are some contributions from excitons to CTEs. However, since the power dependence slope is largely deviated from the germinate recombination with power dependence slope of 1, we can see that the CTEs are mainly formed

through non-germinate recombination of photogenerated carriers. We have added this detailed discussion into the revised manuscript (line 22 on page 7 to line 7 on page 8).

Review comment 4

4. As mentioned above, the surface-trapped excitons do exist close to surface regions, so characterizing the electronic structure at the contact interface between two types of perovskites is of critical importance to understand this phenomenon.

Author response 4

We thank the referee for this important suggestion. We understand the referee's opinion on the electronic structures at the contact interface. We like to mention that there are some difficulties to characterize the electronic structures at the contact interface due to the problem related to spatial resolution. When mixing two types of perovskites (Pb and Sn), the heterostructures are randomly formed within 2D perovskite film. When using our hand-finger pressing method to shear away the Sn perovskite film on the surface of Pb perovskite film, the heterointerfaces are also randomly formed as heterostructured islands. In both situations from mixing and pressing methods, we are facing with a difficulty to spatially resolve the electronic structures at contact interfaces by using electron microscopy techniques. In the future, we hope to use synchrotron technique to solve this problem related to spatial resolution. We thank the referee for this suggestion again.

Here, we like to share our additional information with the referee on the surface-trapped excitons. We observed a weak tail on PL spectrum of pure PEA₂PbI₄ film when the film thickness is reduced (shown in supplementary Fig. S8 below). This weak tail is located around 537.5 nm slightly below the emission (520 nm) from intrinsic excitons, which coincides with the surface-trapped excitons at 537.5 nm shown in transient absorption (Fig. 4a). Basically, this weak tail makes the entire PL spectrum asymmetric. With increasing the film thickness, this weak tail becomes negligible, making the entire PL spectrum symmetric. We believe this weak tail on PL spectrum at thinner film thickness provides a further indication to support the surface-trapped excitons. We have added this detailed discussion into the revised manuscript (line 15 to line 22 on page 12).

In summary, our manuscript presents three critical information related to electronic structures for heterostructures. First, our steady-state and pump-probe transient absorption spectra indicate that the charge-transfer states are formed in excited states under photoexcitation. This means that the charge transfer is occurred between Pb and Sn perovskite structures in excited states. Second, the broad light emission shows that the charge-transfer excitons are 0.48 eV below the band gap (2.33 eV) in the [(PEA₂PbI₄)_{0.999}:(PEA₂SnI₄)_{0.001}] film and 0.17 eV below the bandgap (1.95 eV) in the [(PEA₂SnI₄)_{0.999}:(PEA₂PbI₄)_{0.001}] film. Third, our transient

absorption data indicate that the charge-transfer excitons are quickly formed within 3 ps and generates a light emission in nanoseconds, becoming metastable states.

Supplementary Fig. S8. PL spectra of pure PEA₂PbI₄ films with different thickness. (a) Thickness: ~20 nm; (b) Thickness: ~150 nm.

Review comment 5

5. To corroborate the origin of broad emission in the PEA₂PbI₄/PEA₂SnI₄ heterostructure, the transient absorption for samples formed by using the hand-finger pressing method will help to provide strong pieces of evidence.

Author response 5

We thank the referee for this valuable suggestion. We have measured the transient absorption for our double-layered heterointerfaces prepared by our hand-finger pressing method and added this new result into the revised manuscript (see supplementary Fig. S9 below). We can see that the double-layered heterostructure prepared by our hand-finger pressing method exhibits a broad TA signal between 640 nm and 800 nm in addition to the band-to-band transitions (peaked at 520 nm and 590 nm) in Pb and Sn perovskites. This confirms the formation of CTEs in the double-layered heterostructures prepared by our hand-finger pressing method. We have added this detailed discussion into the revised manuscript (line 6 to line 12 on page 13).

Supplementary Fig. S9. TA spectra at different pump-delay times for double-layered PEA₂PbI₄/PEA₂SnI₄ heterointerfaces prepared by our finger-pressing method.

Review comment 6

6. I am very happy to see the electroluminescence data showing red-shifted broad emission, which is similar to the PL results. In contrast to LEDs, photodetectors may be more suitable for evaluating their practical application.

Author response 6

We thank the referee for the encouraging comment and suggestion. As we can see in our manuscript, in addition to our LED (electroluminescence from charge-transfer excitons), our photodetector also shows that the charge-transfer excitons can be directly excited by a photoexcitation below the bandgap of intrinsic perovskite, extending the photodetection to longer wavelengths.

Review comment 7

7. Minor revisions:

a) There is no space between the number and units (Figure 2 captions), please carefully check similar issues throughout the manuscript.

b) The charge-transfer states at the 2D perovskite/organic interfaces have been discussed in the literature. Please add some discussion in the introduction section.

Author response 7

We thank the referee for the suggestions. We have addressed the issue correspondingly in the revised manuscript and added detailed discussion of the charge-transfer states at the 2D perovskite/organic interfaces in the introduction section (line 12 to 16 on page 4 in the revised manuscript).

Responses to Reviewer #3

Review comment

The manuscript by Hu and co-authors has presented an interesting study on the formation of charge-transfer excitons in quasi-2D perovskite heterostructures. To demonstrate the possibility of CT exciton formation, they used mixed Pb-Sn quasi-2D perovskite as a model system for experimental investigations. The spectrally-broad photoemission from PEA₂PbI₄:PEA₂SnI₄ has been explained in terms of CT excitons, and has been supported by an interesting finger pressing experiment. The CT exciton formation process is studied using transient optical experiments. Overall, the novelty and importance of the paper clearly fall within the scope of Nature Communications. Before recommending publication, the authors are advised to improve their paper by considering the following relatively minor points.

Author response

We thank the referee for the recommendation and suggestions. We have fully considered all review comments and improved our manuscript accordingly in the revised edition.

Review comment 1

1. In the introductory paragraph (page 3), the authors state that: “undoubtedly, the excitons formed within intrinsic perovskite structures are solely responsible for developing high-performance light emitting properties”. The meaning of this statement is rather unclear. Are excitons formed solely responsible for light emitting properties? Apart from recombining radiatively (to emit light) or non-radiatively (to produce heat), excitons may also diffuse and dissociate in a LED device. The authors should consider rewriting this sentence to make it clearer. Also, tuning light emitting properties in hybrid halide perovskite is not experimentally complex. An example of this is the very simple emission color tuning using a mixed halide approach.

Author response 1

Thank the referee for the suggestion. We have revised our writing with more specific means in the introduction session (line 6 to line 11 on page 3 in the revised manuscript). Here is the revised description.

“Undoubtedly, the intrinsic excitons formed within band structures function as the primary excited states responsible for developing high-performance light-emitting properties in such hybrid perovskites. Essentially, tuning the intrinsic excitons through the energy, formation probability, and radiative/nonradiative recombination can determine the light-emitting properties, as exemplified by mixing different halides¹⁰⁻¹², introducing nanostructures¹³⁻¹⁵ and passivating grain boundary defects¹⁶⁻¹⁸.”

Review comment 2

2. *The relatively broad luminescent spectra and the transient optical experiments of the excitations show interesting similarities to what have been reported earlier in Ref 19, where heterostructures formed by (quasi-)2D perovskite, 3D perovskite and polymer were investigated. Ref 19 showed that charge-separated states in the heterostructures were formed within 1 ps before they radiatively recombined more slowly through a bi-molecular process. It was unclear whether the heterostructures were energetically aligned (e.g., type I or type II), but a similar energetic disorder was certainly present. It may be useful for the authors to discuss/comment on these relevant results in relation to the experimental findings of the current paper.*

Author response 2

Thank the referee for the kind suggestion. We have added the further analysis on Ref. 19 into the introduction section in our revised manuscript. Specifically, we acknowledge the Ref.19 in our manuscript that the perovskite-polymer heterostructure demonstrates a broad transient absorption signal below the bandgap that is quickly appeared and slowly lasted into nanoseconds. This presents the possibility that CTEs can be formed through charge transfer between perovskite and organic polymer structures known as high and low-dielectric materials. We further indicate that, in general, CTEs can be formed when a charge transfer is occurred between two adjacent structures with different electron negativities and local energies. We thank the reviewer for asking us to further discuss the Ref. 19. We have added the detailed discussions in the introduction section in the revised manuscript (line 6 to line 12 on page 4).

Review comment 3

3. *To further demonstrate the presence of CTE, the authors measured the photocurrents generated by the perovskite films under sub-bandgap photoexcitation. While this is certainly a nice experiment to do, the reviewer is not 100% convinced that the wavelength of photoexcitation can be considered truly sub-bandgap. The reason for this argument is that the apparent optical bandgap of the PEA₂PbI₄(minority):PEA₂SnI₄(majority) as reported by the authors is 1.95 eV. I presume PEA₂SnI₄ should have very similar (if not the same) bandgap. 1.95 eV is equivalent to a photon wavelength of 635 nm. The authors used a light source with a central wavelength of 640 nm, only 5 nm below the bandgap photon, to provide the photoexcitation. This is very close to the absorption edge of the material. It means that it would not be surprising to observe photocurrent for the semiconductor, as the absorption coefficient at this spectral region so close to the apparent optical bandgap is still not sufficiently close to zero. The type and spectral bandwidth of the light source used are also not specified. The authors should come up with an improved experiment with considerably lower photon energies (much lower than $E_g - kT$). If doing such new experiment is not possible, the authors should comment on the potential weakness of the relevant conclusion drawn from the current experimental setup.*

Author response 3

We thank the referee for the comment and suggestion. We understand that using a photoexcitation with the energy further below the bandgap would be a better choice to directly excite the CTEs in our photocurrent studies.

In order to exclude the possible light absorption from PEA_2SnI_4 perovskite, we tested the photo response behavior of the relevant device with only pure PEA_2SnI_4 as active layer (Fig. 5a in the revised manuscript). As shown in the figure that nearly no photocurrent is generated (ON/OFF ratio = 3.91) under the illumination of 640 nm laser. On the contrary, device with heterostructures show ON/OFF ratio of 532. It verifies that the photocurrent in latter device are generated from CTEs rather than from PEA_2SnI_4 perovskite. We have revised the Fig. 5a and added further discussion (line 9 to line 11 on page 14 in the revised manuscript).

Fig. 5a. The ON-OFF behaviors of $[(\text{PEA}_2\text{PbI}_4)_{0.95}:(\text{PEA}_2\text{SnI}_4)_{0.05}]$ (red) photodetector as compared to pure PEA_2PbI_4 (black) and PEA_2SnI_4 (blue) devices when exciting at 640 nm ($1000\text{mW}/\text{cm}^2$).

References

1. Verhoeven, J. W., van Ramesdonk, H. J., Groeneveld, M. M., Benniston, A. C. & Harriman, A. Long-lived charge-transfer states in compact donor–acceptor dyads. *Chem. Phys. Chem.* **6**, 2251-2260 (2005).
2. Deotare, P. B. et al. Nanoscale transport of charge-transfer states in organic donor–acceptor blends. *Nature Mater.* **14**, 1130-1134 (2015).
3. Loi, M. A. et al. Charge transfer excitons in bulk heterojunctions of a polyfluorene copolymer and a fullerene derivative. *Adv. Funct. Mater.* **17**, 2111-2116 (2007).
4. Zhu, T. et al. Highly mobile charge-transfer excitons in two-dimensional WS₂/tetracene heterostructures. *Sci. Adv.*, **4**, eaao3104 (2018).
5. Bettis Homan, S. et al. Ultrafast exciton dissociation and long-lived charge separation in a photovoltaic pentacene–MoS₂ van der Waals heterojunction. *Nano Lett.* **17**, 164-169 (2017).
6. Hong, X. et al. Ultrafast charge transfer in atomically thin MoS₂/WS₂ heterostructures. *Nat. Nanotechnol.*, **9**, 682 (2014).
7. Wu, K. et al. Efficient and ultrafast formation of long-lived charge-transfer exciton state in atomically thin cadmium selenide/cadmium telluride type-II heteronanosheets. *ACS Nano* **9**, 961-968 (2015).
8. Liang, D. et al. Color-pure violet-light-emitting diodes based on layered lead halide perovskite nanoplates. *ACS Nano* **10**, 6897-6904 (2016).
9. Zhu, X. et al. Vapor-fumigation for record efficiency two-dimensional perovskite solar cells with superior stability. *Energy & Environ. Sci.* **11**, 3349 (2018).
10. Yoon, H.C. et al. Study of perovskite QD down-converted LEDs and six-color white LEDs for future displays with excellent color performance. *ACS Appl. Mater. Interfaces* **8**, 18189-18200 (2016).
11. Yoon, Y.J. et al. Full-color luminescence by post-treatment of perovskite nanocrystals. *Joule* **2**, 2105-2116 (2018).
12. Smith, M.D., Connor, B.A. & Karunadasa, H.I. Tuning the luminescence of layered halide perovskites. *Chem. Rev.* **119**, 3104-3139 (2019).
13. Hassan, Y. et al. Structure-tuned lead halide perovskite nanocrystals. *Adv. Mater.* **28**, 566-573 (2016).

-
14. Mao, J. et al. Novel direct nanopatterning approach to fabricate periodically nanostructured perovskite for optoelectronic applications. *Adv. Funct. Mater.* **27**, 1606525 (2017).
 15. Ban, M. et al. Solution-processed perovskite light emitting diodes with efficiency exceeding 15% through additive-controlled nanostructure tailoring. *Nature Commun.* **9**, 3892 (2018).
 16. Lin, K. et al. Perovskite light-emitting diodes with external quantum efficiency exceeding 20 per cent. *Nature* **562**, 245-248 (2018).
 17. Wang, H. et al. Trifluoroacetate induced small-grained CsPbBr₃ perovskite films result in efficient and stable light-emitting devices. *Nature Commun.* **10**, 665 (2019).
 18. Qin, J. et al. Enabling self-passivation by attaching small grains on surfaces of large grains toward high-performance perovskite LEDs. *iScience* **19**, 378-387 (2019).

Reviewers' Comments:

Reviewer #1:

Remarks to the Author:

The authors have addressed my comments adequately. I suggest the authors to move the inset from the revised Fig. 1S out of the figure, and plot it as an independent panel, as it covers part of the features in the XRD.

Reviewer #2:

Remarks to the Author:

In the revised manuscript, concerns raised by the reviewer have been addressed very well, and explanations about some observations are convincing. The paper could be accepted for publication in Nature Communications without any changes.

Reviewer #3:

Remarks to the Author:

The authors have addressed my review comments in a satisfactory manner. The revised paper is now suitable for publication in Nature Communications.

Responses to Reviewer #1

Review comment

The authors have addressed my comments adequately. I suggest the authors to move the inset from the revised Fig. 1S out of the figure, and plot it as an independent panel, as it covers part of the features in the XRD.

Author response

We thank the referee for the recommendation and suggestion. In the revised manuscript, we have moved the inset of Supplementary Fig.1 out as Supplementary Fig.1b.

Supplementary Fig. 1 | X-ray diffraction (XRD) patterns obtained from different films. **a** All five films show an in-plane orientation of PbI_6 or SnI_6 sheets along substrate surface. **b** Zoom-in data to show intensities and peak positions for all five films. Blue curve: PEA_2PbI_4 , sky blue curve (overlapped with green curve): $[(\text{PEA}_2\text{PbI}_4)_{0.99};(\text{PEA}_2\text{SnI}_4)_{0.01}]$, green curve: $[(\text{PEA}_2\text{PbI}_4)_{0.95};(\text{PEA}_2\text{SnI}_4)_{0.05}]$, orange curve: $[(\text{PEA}_2\text{PbI}_4)_{0.05};(\text{PEA}_2\text{SnI}_4)_{0.95}]$ and red curve: PEA_2SnI_4 .

Responses to Reviewer #2

Review comment

In the revised manuscript, concerns raised by the reviewer have been addressed very well, and explanations about some observations are convincing. The paper could be accepted for publication in Nature Communications without any changes.

Author response

We thank the referee for the recommendation and suggestion.

Responses to Reviewer #3

Review comment

The authors have addressed my review comments in a satisfactory manner. The revised paper is now suitable for publication in Nature Communications.

Author response

We thank the referee for the recommendation and suggestion.